# The Lions Sports Academy Tackle*TEK* Tool: The intra-and inter-coach reliability of assessing tackling competency in rugby union

Robert Owen[1]*, Camilla J. Knight[1,5,7], Will Page[2], Jamie Tallent[3,4], Liam Kilduff[1,5], Mark Waldron[1,5,6]

1 Applied Sports, Technology, Exercise and Medicine (A-STEM), Swansea University, Swansea, United Kingdom, 2 Faculty of Sport, Technology and Health Sciences, St Mary's University, Twickenham, United Kingdom, 3 School of Sport, Rehabilitation, and Exercise Sciences, University of Essex, Colchester, United Kingdom, 4 Monash University Exercise Neuroplasticity Research Unit, School of Primary and Allied Care, Monash University, Frankston, Australia, 5 Welsh Institute of Performance Science, Swansea University, Swansea, United Kingdom, 6 School of Health and Behavioural Sciences, University of the Sunshine Coast, Queensland, Australia, 7 Department of Physical Education and Sport Science, University of Agder, Norway

☉ These authors contributed equally to this work.
* 2029603@swansea.ac.uk

## Abstract

Tackling skill tests in rugby aim to evaluate skill competency, but their reliability to detect meaningful changes in tackling competency over time remains uncertain. This study aimed to: 1) determine the intra- and inter-rater reliability of scoring composite tackle competency by three youth grassroots rugby coaches (tier one, two, and three), and 2) assess reliability in scoring across tackling subcomponents (pre-contact, contact, and post-contact) using the Tackle*TEK* Tool. Pilot data was used to develop *a-priori* analytical goals to assess the reliability of composite and sub-component scoring. Non-parametric 95% Limits of Agreement were used to assess reliability. Twenty-five participants were recorded performing the one *vs.* one tackling assessment. The three coaches scored each tackle against pre-defined 18-point criteria. Systematic differences were assessed using Wilcoxon tests. Intra-coach composite tackling scores demonstrated agreement levels of 72% (95% CI 60% to 84%) for the tier three coach, and 100% (95% CI 100% to 100%) for both the tier two and one coaches, relative to the analytical goal. Inter-coach composite tackling agreement ranged from 78% (95% CI 67% to 89%) to 100% (95% CI 100% to 100%). Intra-coach subcomponent analysis showed the tier one and tier two coach to score all tackling subcomponents consistently (*P* > 0.05); however, the tier three coach only reliably scored the pre-contact phase. Subcomponent agreement ranged from 68% (95% CI 55% to 81%) to 82% (95% CI 71% to 93%), between tier three and tier one coaches, to 72% (95% CI 60% to 84%) to 98% (95% CI 94% to 100%), between tier two and tier one coaches. Within the context of this study, agreement was strongest

**Data availability statement:** The data for this research is held in an open repository (10.5281/zenodo.16740441).

**Funding:** This work was funded by Lions Sports Academy Ltd (https://lionssports.academy/). The grant (AMS106867) was awarded to MW. The funders facilitated data collection through collaboration with Swansea University.

**Competing interests:** Lions Sports Academy awarded MW a grant (AMS106867) for the research.

for tier one and tier two coaches, supporting the use of the Tackle*TEK* Tool to assess changes in tackling competency over time, with further work required to extend its application across coaching levels.

## Introduction

Skill testing among youth team athletes is common practice in talented player pathways [1–3]. Results from these tests are currently used to differentiate between playing levels in research [1,4,5], or in practice, to establish an athlete's competency in executing a sport-specific skill [1,6,7]. These tests often include an objective scoring outcome, such as an accuracy score [7,8] or time to complete a given task [9], with the results used to demonstrate the efficacy of a training cycle [2,10] or establish players' readiness to train [11].

In rugby, various skill assessments, such as passing [12], have been adopted but it has been historically challenging to evaluate the competency of tackling skills. Assessment of tackling skills are more problematic, since they encompass interaction between players and, therefore, scores may alter depending on the ability of the defender or ball-carrier. This means that tackling may be more inconsistently performed and subject to variability in execution during an assessment [13,14]. Thus, in a testing scenario, consistent visual recognition of the correct tackling skill is reportedly compromised in the assessment of elite youth rugby league players [15]. This issue could be exacerbated among youth players, whose movement skills can be more variable [16,17], increasing the difficulty of characterising more advanced skills, such as tackling. These issues are unfortunate, as it is vital that young players execute tackling technique correctly. Indeed, the success of tackling outcomes [18,19] and, more importantly, the safety of the tackling players [20–22] is reliant upon the correct tackling skill execution.

One approach to the assessment of tackling has been to use rugby coaches' interpretation to score the skilled performance according to pre-set criteria, written by coaches and experts in the field [11,15]. Typically, scoring personnel (live scoring) or cameras (delayed scoring) are used to rate one *vs.* one tackles [2,11,23]. Criteria have included elements of the tackle approach (pre-contact), tackle execution (contact), and tackle completion (post-contact) [2,15,24], although some assessments focus solely on criteria during the contact phase [11,13]. However, amid growing player safety concerns, numerous risk factors (i.e., more likely to result in injury) have been identified during tackle events, such as technical breakdown (i.e., initial contact points; [25]), height of the tackle impact, speed of the collision, and fatigue [25,26]. These detailed reports have improved the understanding of tackling biomechanics [25,27,28], which provide specific coaching feedback to mitigate tackle-associated injury risk. Thus, criteria for safe tackling technique can be updated and incorporated into new tackling assessments.

It is vital that the intra- and inter-rater reliability (i.e., error within and between raters, respectively) of tackling assessments that require coach interpretation is

first quantified to enable coaches to determine its ability to detect improvements in tackling that might occur across time among players. The intra- and inter-rater reliability of novice and expert raters has been reported in youth rugby league [11,15]. In one of these studies [15], a non-parametric equivalent of the 95% limits of agreement (NP 95% LOA) was adopted [29], and analytical goals were targeted using methods detailed by Atkinson & Nevill [30]. For data on nominal or ordinal scales, such as video data recorded during coaching assessments, using NP 95% LOA is more appropriate, but more importantly, offers a more robust statistical method than correlation-based indices such as the intraclass correlation coefficient (ICC), as it quantifies absolute agreement rather than associations between repeated measurement ranks [30,35] The NP 95% LOA method quantifies measurement error by capturing both systematic and random error, and aligns with context-specific analytical goals, unlike methods such as ICC [30]. It also shows the range within which 95% of the differences are expected to occur between repeated tests. These results can then be compared against predefined thresholds, i.e., context-specific analytical goals, to interpret whether the measurement error is acceptable for real-word use [30]. A new tackle assessment model for youth rugby union has been developed by Lions Sports Academy – 'The Tackle*TEK* Tool'. The Tackle*TEK* Tool standardised protocol and criteria were co-developed with academics and grassroots coaches, the intended users of the assessment, consistent with previous coach-led skill assessments [11,15]. Assessment development followed a structured workflow that included: 1) identification of tackling competency indicators for grassroots coaches, 2) identification of tackling competency indicators in academic literature, 3) alignment of these indicators with coach-understandable criteria, 4) pilot testing specifically within a grassroots environment, and 5) refinement of the protocol and scoring based on feasibility and clarity for interpretation by grassroots coaches. The Tackle*TEK* Tool was designed to provide numerical scoring of tackling competency as part of the Tackle*TEK* Programme. To ensure that reliability was evaluated in relation to meaningful applied change, Atkinson and Nevill [30] recommend the use of analytical goals derived from empirical evidence rather than arbitrary statistical thresholds. Such goals provide an objective, data-driven basis for defining a realistic, real-world threshold of meaningful change, mitigating reliance on subjective judgement [30]. In this context, analytical goals refer to *a-priori* thresholds of change magnitude considered meaningful for practical interpretation, rather than individual scoring criteria within the assessment tool.

Prior to the present reliability study, pilot data were collected to examine changes in tackling competency following implementation of the Tackle*TEK* programme in a grassroots rugby context. Youth players were assessed using the same 18-point composite tackling competency scoring framework described in the present manuscript, with the primary outcome being change in composite score across the intervention period.

Pilot data demonstrated a median improvement of four arbitrary units (AU; maximum score of 18 AU) in the composite tackling competency score following five weeks of the Tackle*TEK* programme [31]. Accordingly, an analytical goal of four AU was defined as the minimum change in the composite score considered meaningful and was used to contextualise measurement error when evaluating intra- and inter-coach agreement. Arbitrary units were used owing to no standardised unit for the measurement of tackling competency. Therefore, in accordance with recommendations [30], in the current study a change equal to or less than four AU was determined as the *a-priori* analytical goal for composite tackling scores. To understand the rating of composite tackling skill execution, subcomponents of the tackling skill were defined as the pre-contact, contact, and post contact phase in keeping with current research. Subcomponent analytical goals were also identified from pilot data as: 'pre-contact phase' of two AU (of nine available units), the 'contact phase' of one AU (of five available units), and the 'post-contact phase' of one AU (of four available units). That is, for the Tackle*TEK* Tool to be sufficiently sensitive to identify a change in tackle skill competency across time, the error within and between coaches rating players' competency must not exceed these pre-identified thresholds. Thus, the aim of this study was twofold: 1) to determine the intra- and inter-coach reliability (i.e., agreement) across a tier one, tier two, and tier three coach while using the Tackle*TEK* Tool and 2) to assess agreement in scoring across three subcomponents (pre-contact, contact and post-contact) of the tackle.

## Methods

### Participants

Written informed consent and parental assent was obtained from 25 participants (range: age 9–14 years; body mass 27.8 to 79.5 kg; stature 137.1 to 187.4 cm) to participate in the study. All participants (i.e., the school children) had two academic years of full-contact rugby union training where participants were regularly coached rugby by qualified coaches, weekly (180−240 minutes) for twelve weeks throughout the school term participating in regular grassroots inter-school match play. All data collection was performed during the school term from September to April outdoors, between the hours of 2−4 pm, as part of regular games lessons. Institutional ethical approval was provided for this study (Swansea University FSE Research Ethics Committee; RO_25–10-22b). Data was collected between 08/02/22 and 01/10/2024.

### Design

An inter- and intra-coach reliability assessment of the 'Tackle*TEK* Tool' was conducted. Participants were initially familiarised with the protocol by emulating the one *vs.* one tackling assessment once, limiting the learning effect. Participants then performed the Tackle*TEK* assessment a week later. The one *vs.* one rugby tackling assessment was influenced by previous research that adopted similar procedures with professional and academy rugby players, e.g., the assessment grid size [11,15], and the competency criteria [11,15,32] which the current study updated with current research of effective tackling biomechanics [27,28,33].

The rugby tackle assessment was performed in a 10 x 10 m grid. Three cameras were fixed into position 1 m from the ground and 10 m from the perimeter of the grid to record the tackle event from the frontal and sagittal planes (Fig 1). Two frontal plane cameras recorded the tackle event from the front and rear of the defender and the sagittal plane camera captured the tackle event from the side. Ball-carriers were instructed to '*jog forward towards the try line and be prepared to be tackled*'. This was to provide the defender the best opportunity to make a tackle. The pace was self-determined by participants, as the assessment is intended for youth players, varying in confidence and ability. The instruction given to the

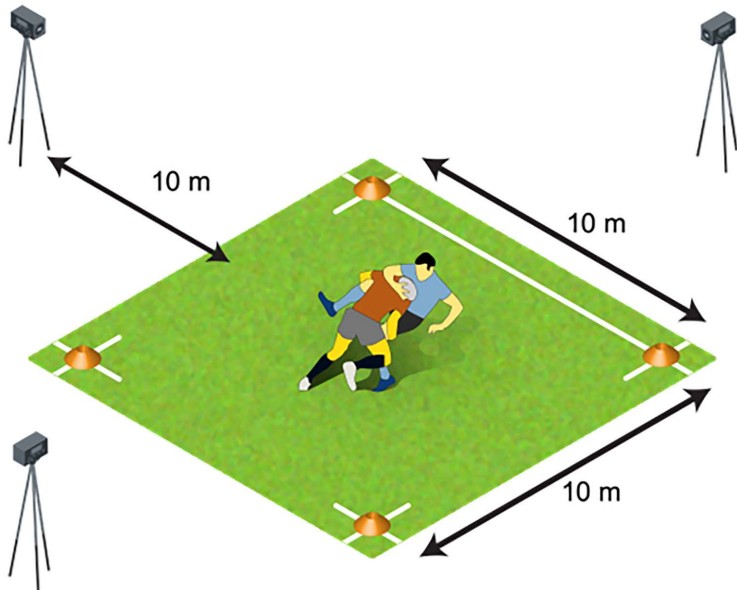

**Fig 1. The environment and camera set up for the Lions Sports Academy tackling competency assessment.**

tackler was to '*make a safe and effective tackle*'. The only requirement for participant pairing was being in the same age-grade. As per current tackling assessments, tackling criteria were co-developed by university academics and coaches, using previous research [11,15,32] updated with current biomechanics research [27,28,33] to confirm the 18-point criteria. Grassroots coaches supported the knowledge translation of the criterion to promote understanding by the user (grassroots rugby coaches). The tackle assessment video recordings were used to analyse the participant rugby tackle competency against 18-point criteria (Table 1), with three raters (rugby coaches) of varying qualifications and experience.

## Familiarisation of raters

Player tackling competency was assessed by three qualified coaches, which were labelled as 'tier one', 'tier two', and 'tier three' for the purposes of the current study, based on their experience and qualification level (Table 2). On the day of the first assessment, preceding the first tackling assessment, the lead researcher (RO) conducted a familiarisation session with the raters. In this session, the criteria were defined, explained, and agreed upon. Each criterion was explained with

**Table 1. Eighteen-point criteria to assess rugby tackler technical skill competency using the Lions Sports Academy Tackle*TEK* Tool, across three components.**

| *Component 1: Pre-contact* |
| --- |
| Step into the ball-carriers base of support |
| Track the ball-carrier onto the shoulder |
| Transition from a high to low body position |
| Maintain a straight spine position |
| Centre of gravity ahead of the base of support |
| Body position square to the ball-carrier |
| Adopt a 'boxer stance' with the arms |
| Head up, looking forward beyond the ball-carrier |
| Shortening of steps as the defender approaches the ball-carrier |
| *Component 2: Contact* |
| Explode forward on contact |
| Initial contact with the shoulder |
| Same lead leg as the shoulder making contact with the ball-carrier |
| Contact height |
| Head in a safe position |
| *Component 3: Post-contact* |
| Arms punch forward and wrap around the ball-carrier |
| Active leg drive |
| Body finishes on top of the ball-carrier |
| Head finishes on top of the ball-carrier |

**Table 2. Description of coach classification based upon rugby coaching qualification and number of years coaching youth rugby union.**

| Coach tier | Years coaching youth rugby union | Highest level of ruby coaching qualification |
| --- | --- | --- |
| Tier three | 32 | England Rugby Advanced Coaching Award |
| Tier two | 15 | England Rugby Coaching Award |
| Tier one | <1 | England Rugby Coaching Award |

visual assistance and demonstration using a tackle event video. The demonstration video was not included in the reliability assessment. During the familiarisation session, coaches had the opportunity to ask questions in isolation of other raters. The first assessment was completed in isolation following the familiarisation session.

### Video selection

Twenty five of 206 tackling videos were randomly selected from tackling assessments recorded as part of the Tackle*TEK* programme in 2022 and 2023. Qualifying participant video criteria were: i) healthy participants who were actively participating in school rugby training, ii) participating in contact rugby union in school for at least two years, and iii) had no prior relationship to the coaches rating tackling competency. The final 25 videos were randomly selected from 206 tackle videos which met the qualifying criteria. Each video was allocated an identification number and numbers were randomly generated (www.random.org) to select the tackles for assessment. The three raters had not previously watched nor assessed the final 25 video recordings of the rugby tackles. All videos were recorded with the same multi-camera setup (Sony HDR-PJ810E Handycam, 50 Hz, Singapore) and recorded at 30 frames per second.

## Procedures

### Video editing

The final randomly selected 25 tackling videos were edited using Microsoft Clipchamp (Clipchamp, version 2.9.1.0, Microsoft, Australia). The edited footage showed the tackle event from three camera angles (frontal plane, front and rear view of the tackler; sagittal plane, side view of the tackler), as done by Gabbett [11] and Gabbett and Kelly [23]. The same 25 tackle videos were also edited to be assessed from a single, sagittal plane, camera angle (sagittal plane, side view of the tackler), in line with Waldron et al. [15] and den Hollander et al. [32]. In total, 50 rugby tackle videos were produced for rater assessment, comprising 25 individual tackle events, repeated twice, observed from different camera angles, that rendered a sample greater than 40, that is recommended by Altman [34].

### Coaches and assessment

The coaches coded the 50 tackle videos twice, once on each of the two occasions, one week apart. Coaches could play, pause, and rewind each tackle video to perform the analysis. Rugby tackle skill competency was assessed against 18-point criteria (Table 1). Each individual criterion measure was worth one point. The coaches would score one point for each criterion successfully identified and a score of 0 was given if the criterion was not met. The pre-contact, contact, and post-contact subcomponents were worth nine, five, and four points, respectively. Each coach was blind to the other coaches' responses, their own previous responses on retest (unable to review previously scored tackle ratings), as well as the age, and playing level of the participants in the videos.

### Statistical analysis

Wilcoxon tests were used to establish intra-coach reliability. Non-parametric Kruskal-Wallis tests were used to test for differences (i.e., systematic bias) in the composite and subcomponent (pre-contact, contact, and post-contact) scores from the Tackle*TEK* Tool between coaches (tier three, tier two, and tier one). Post-hoc Mann Whitney-U tests were used for pairwise comparisons of the coaches. The reliability was assessed using a non-parametric 95% limits of agreement (NP 95% LOA; [29,35]). As established, a reference value of four points on the Tackle*TEK* Tool or less was considered to be important to detect sensitivity to change in tackle competency over time. An analysis of 'perfect agreement' was also performed to understand the errors within and between coaches when all tolerances are removed as if the tool was intended for the purpose of talent identification in line with that of previous assessments [15]. Statistical significance across all tests was determined as $P < 0.05$ for intra-rater analysis, with statistical analyses conducted on IBM SPSS Statistics (version

29.0). Adjustments to the alpha level were conducted using the Bonferroni correction, within each hypothesis, to the *P* value to reduce type I error rate for inter-rater analysis. The correction formula consists of dividing the α = alpha level (0.05) by the number of comparisons (*n*) made within each hypothesis. Therefore, α/n would be the new critical *P* value (0.017) when conducting a three-way comparison. Data are reported as medians and inter-quartile ranges (IQR).

## Results

### Intra-coach reliability assessment

**Composite tackling skill competency scores.** Wilcoxon tests revealed that, within raters, the tier two (Z = −0.894, *P* = 0.371) and tier one (Z = −0.264, *P* = 0.792) coaches demonstrated no systematic bias between the first and second assessment (Table 3). Both the tier two and tier one coaches achieved the *a-priori* analytical goal of four or less at a 100% tier, and achieved 70% and 86% of perfect agreement, respectively (Table 4). However, the tier three coach systematically scored tackling competency higher in the second assessment compared with the first (Z = −3.876, *P* ≤ 0.001; Table 3). Also, the tier three coach demonstrated weaker levels of agreement, scoring 72% agreement (95% CI 60% to 84%) in relation to the analytical goal. The tier two (Z = −0.894, *P* = 0.371) and tier one (Z = −0.264, *P* = 0.792) coaches both repeated the tackle competency assessment without systematic bias identified between the first and second assessment (Table 3).

### Subcomponent tackling skill competency scores

Wilcoxon tests revealed, for the tier three coach, no difference in tackling scores in pre-contact (Z = −1.482, *P* = 0.138); however, differences were identified in the contact (Z = −4.159, *P* ≤ 0.001) and post-contact (Z = −4.302, *P* ≤ 0.001) phases (Table 3). The tier three coach scored tackling competency to 64% (95% CI 51% to 70%), 68% (95% CI 55% to 81%), and

**Table 3. The median and inter-quartile range for the tackling assessment of the intra-coach composite and subcomponent competency score across two qualified experienced rugby union coaches and a qualified inexperienced rugby union coach.**

| | | | Assessment one | | | Assessment two | | |
| | | | Composite score | | | Composite score | | |
| | Z-score | P-value | Median | 25th Percentile | 75th Percentile | Median | 25th Percentile | 75th Percentile |
|---|---|---|---|---|---|---|---|---|
| Tier three coach | −3.867 | ≤ 0.001 | 8 | 5 | 11 | 10 | 7 | 13 |
| Tier two coach | −0.894 | 0.371 | 12 | 10 | 13 | 12 | 11 | 13 |
| Tier one coach | −0.264 | 0.792 | 11 | 10 | 13 | 11 | 10 | 13 |
| | | | Pre-contact | | | Pre-contact | | |
| Tier three coach | −1.482 | 0.138 | 4 | 2 | 5 | 4 | 3 | 6 |
| Tier two coach | −1.069 | 0.285 | 5 | 4 | 6 | 5 | 4 | 6 |
| Tier one coach | 0.00 | 1.000 | 6 | 5 | 6 | 6 | 5 | 6 |
| | | | Contact | | | Contact | | |
| Tier three coach | −4.159 | ≤0.001 | 3 | 2 | 4 | 3 | 3 | 5 |
| Tier two coach | 0.000 | 1.000 | 4 | 3 | 4 | 4 | 3 | 4 |
| Tier one coach | 0.000 | 1.000 | 4 | 3 | 5 | 4 | 3 | 5 |
| | | | Post-contact | | | Post-contact | | |
| Tier three coach | −4.302 | ≤0.001 | 1 | 0 | 2 | 3 | 2 | 3 |
| Tier two coach | −0.447 | 0.655 | 3 | 2 | 3 | 3 | 2 | 3 |
| Tier one coach | −1.000 | 0.317 | 2 | 1 | 2 | 2 | 1 | 2 |

[a]A P-value of < 0.017 indicates a systematic difference in tackling scores.

**Table 4. Non-parametric 95% limits of agreement for intra-coach composite tackling competency scores with an analytical goal of perfect agreement and an analytical goal of four or less.**

| | Perfect agreement | | 4 or less | |
|---|---|---|---|---|
| | PA±0 | 95% CIs | PA≤4 | 95% CIs |
| | (%) | | (%) | (%) |
| Tier three coach | 12 | 3–21 | 72 | 60–84 |
| Tier two coach | 70 | 57–83 | 100 | 100–100 |
| Tier one coach | 86 | 76–96 | 100 | 100–100 |

[a]PA: perfect agreement, CIs: Confidence intervals.

80% (95% CI 69–81%) level of agreement in relation to the analytical goal for the pre-contact, contact and post-contact phases, respectively (Table 5). The tier two coach showed no difference scoring the pre-contact ($Z=-1.069$, $P=0.285$), contact ($Z=0.000$, $P=1.000$), and post-contact ($Z=-0.447$, $P=0.655$) phases (Table 3). The tier one coach also demonstrated no difference scoring the pre-contact ($Z=0.000$, $P=1.000$), contact ($Z=0.000$, $P=1.000$), and post-contact phases ($Z=-1.000$, $P=0.317$) (Table 3). Both, the tier two and tier one coach achieved 100% (95% CI 100% to 100%) agreement across all subcomponents in relation to the analytical goals (Table 5).

### Inter-coach reliability assessment

**Composite tackling skill competency score.** Kruskal-Wallis tests demonstrated differences across all coaches in the first tackling assessment ($X^2_{(2)} = 31.859$, $P\leq0.001$) but not in the second assessment ($X^2_{(2)} = 2.893$, $P=0.235$). Pairwise analysis demonstrated no statistical differences in tackling competency scores between the tier two and tier one coach across assessment one ($P=0.627$), but the tier three coach scored lower composite tackling scores than both the tier one coach ($P\leq0.001$) and the tier two coach two ($P\leq0.001$) during the first assessment. Based upon the composite score analytical goal of four or less points, the tier three coach demonstrated the weakest level of agreement, which improved from 62% (*vs.* the tier one coach) and 70% (*vs.* the tier two coach) in assessment one to 78% (*vs.* both raters) in assessment two (Table 6). Tier two and tier one coaches achieved 100% inter-coach agreement within the analytical goal in both the first and second assessments (Table 6).

### Subcomponent tackling skill competency scores

When scoring the tackling competency subcomponents (pre-contact, contact, and post-contact), Wilcoxon tests showed the tier three coach to score repeatably in both assessments one and two during the pre-contact phase ($Z=-1.482$, $P=0.138$) (Table 3), but could not accurately repeat scoring during the contact ($Z=-4.159$, $P\leq0.001$) and post-contact ($Z=-1.302$, $P\leq0.001$) phases (Table 3). The tier two coach was able to accurately repeat the scoring of tackling competency in the pre-contact ($Z=-1.069$, $P=0.285$), contact ($Z=0.000$, $P=1.000$), and post-contact ($Z=-0.447$, $P=0.655$) subcomponents (Table 3). The tier one coach also demonstrated accurate repeat scoring of tackling competencies across all subcomponents, pre-contact ($Z=0.000$, $P=1.000$), contact ($Z=0.000$, $P=1.000$), and post-contact ($Z=-1.000$, $P=0.317$) (Table 3).

Kruskal-Wallis tests revealed systemic differences between the three coaches in the pre-contact ($X^2_{(2)} = 25.939$, $P\leq0.001$), contact ($X^2_{(2)} = 26.493$, $P\leq0.001$), and post-contact ($X^2_{(2)} = 23.145$, $P\leq0.001$) phases during the first assessment. Pairwise analysis of assessment one revealed differences between the tier three and tier two coach at the pre-contact ($P\leq0.001$) and contact ($P\leq0.001$) phases; and differences between the tier three and tier one coach in all subcomponent phases ($P\leq0.001$). There are differences observed between the tier two and tier one coach in the post-contact phase ($P\leq0.001$). In assessment two, Kruskal-Wallis tests demonstrate no systemic differences between

**Table 5. Non-parametric 95% limits of agreement for intra-coach pre-contact, contact, and post contact subcomponent tackling competency scores with an analytical goal of perfect agreement for all subcomponents, and specific analytical goals of two or less for pre-contact, one or less for contact, and one or less for post-contact.**

| | Pre-contact | | | |
|---|---|---|---|---|
| | **Perfect agreement** | | **2 or less** | |
| | PA±0 | 95% CIs | PA≤2 | 95% CIs |
| | (%) | (%) | (%) | (%) |
| Tier three coach | 20 | 9–31 | 64 | 51–77 |
| Tier two coach | 84 | 74–94 | 100 | 100–100 |
| Tier one coach | 80 | 69–91 | 100 | 100–100 |
| | **Contact** | | | |
| | **Perfect agreement** | | **1 or less** | |
| | PA±0 | 95% CIs | PA≤1 | 95% CIs |
| | (%) | (%) | (%) | (%) |
| Tier three coach | 28 | 16–40 | 68 | 55–81 |
| Tier two coach | 100 | 100–100 | 100 | 100–100 |
| Tier one coach | 100 | 100–100 | 100 | 100–100 |
| | **Post-contact** | | | |
| | **Perfect agreement** | | **1 or less** | |
| | PA±0 | 95% CIs | PA≤1 | 95% CIs |
| | (%) | (%) | (%) | (%) |
| Tier three coach | 32 | 19–45 | 80 | 69–81 |
| Tier two coach | 96 | 91–100 | 100 | 100–100 |
| Tier one coach | 100 | 100–100 | 100 | 100–100 |

[a]PA: perfect agreement, CIs: Confidence intervals.

**Table 6. Comparisons of the inter-coach composite tackling competency scores between two qualified experienced coaches and a qualified inexperienced coach.**

| | P-value | Perfect agreement | | 4 or less | |
|---|---|---|---|---|---|
| | | PA±0 | 95% CIs | PA≤4 | 95% CIs |
| | | (%) | | (%) | (%) |
| | | Assessment one | | | |
| Tier three coach *vs.* tier two coach | ≤ 0.001 | 6 | −1–13 | 70 | 57–83 |
| Tier three coach *vs.* tier one coach | ≤ 0.001 | 8 | 0–16 | 62 | 49–75 |
| Tier two coach *vs.* tier one coach | 0.206 | 28 | 16–40 | 100 | 100–100 |
| | | Assessment two | | | |
| Tier three coach *vs.* tier two coach | 0.024 | 6 | −1–13 | 78 | 67–89 |
| Tier three coach *vs.* tier one coach | 0.106 | 8 | 0–16 | 78 | 67–89 |
| Tier two coach *vs.* tier one coach | 0.074 | 32 | 19–45 | 100 | 100–100 |

a PA: perfect agreement, CIs: Confidence intervals.

b A P-value of < 0.017 indicates a systematic difference in tackling scores.

coaches in the pre-contact ($X^2_{(2)}$ = 7.385, $P$ = 0.025) and contact ($X^2_{(2)}$ = 2.821, $P$ = 0.244) phases but reveal a difference in scoring of the post-contact ($X^2_{(2)}$ = 15.453, $P \leq 0.001$) phase. Pairwise analysis of assessment two reveal the differences to be between the tier three and tier two coach ($P$ = 0.009) and between the tier three and tier one coach ($P \leq 0.001$). The tier three and tier two coach demonstrated a 72% (95% CI 60% to 84%) and 70% (95% CI 57% to 83%) level of agreement, in scoring the pre-contact phase in relation to the analytical goal, during assessment one and two, respectively (Table 7). The tier three and tier two coach improved their level of agreement, in accordance with the *a-priori* goal, from assessment one to assessment two in both the contact phase, from 70% (95% CI 57% to 83%) to 86% (95% CI 76% to 96%), and the post-contact phase from 62% (CI 49% to 75%) to 74% (95% CI 62% to 86%) (Table 7). The tier three and tier one coach improved their level of agreement across all subcomponents; in pre-contact phase from 68% (95% CI 55% to 81%) to 70% (95% CI 57% to 83%); in the contact phase from 64% to 82% (95% CI 51% to 77%); and in the post contact phase from 62% (95% CI 49% to 75%) to 68% (95% CI 55% to 81%). The tier two and tier one coach agreed to a tier of 98% (95% CI 94% to 100%), 92% (95% CI 84% to 100%), and 72% (95% CI 60% to 84%) across the pre-contact, contact, and post-contact phases, respectively, in relation to the analytical goals, in both assessments.

## Discussion

The present study aimed to, firstly, determine the intra- and inter-coach reliability of the new Tackle*TEK* Tool. The second aim was to assess the agreement of tackling competency scoring in the three subcomponents (pre-contact, contact, and

**Table 7. Comparison of the inter-coach subcomponent tackling competency scores between two qualified experienced coaches and a qualified inexperienced coach with analytical goals of perfect agreement for all subcomponents, and specific analytical goals of two or less for pre-contact, 1 for contact, and 1 for post-contact.**

| | Assessment one | | | | Assessment two | | | |
|---|---|---|---|---|---|---|---|---|
| | **Pre-contact** | | | | | | | |
| | **Perfect agreement** | | **2 or less** | | **Perfect agreement** | | **2 or less** | |
| | **PA±0** | **95% CIs** | **PA≤2** | **95% CIs** | **PA±0** | **95% CIs** | **PA≤2** | **95% CIs** |
| | **(%)** | **(%)** | **(%)** | **(%)** | **(%)** | **(%)** | **(%)** | **(%)** |
| Tier three coach *vs.* tier two coach | 16 | 6–26 | 72 | 60–84 | 18 | 7–29 | 70 | 57–83 |
| Tier three coach *vs.* tier one coach | 16 | 6–26 | 68 | 55–81 | 16 | 6–26 | 70 | 57–83 |
| Tier two coach *vs.* tier one coach | 32 | 19–45 | 98 | 94–100 | 30 | 17–43 | 98 | 94–100 |
| | **Contact** | | | | | | | |
| | **Perfect agreement** | | **1 or less** | | **Perfect agreement** | | **1 or less** | |
| | PA±0 | 95% CIs | PA≤1 | 95% CIs | PA±0 | 95% CIs | PA≤1 | 95% CIs |
| | (%) | (%) | (%) | (%) | (%) | (%) | (%) | (%) |
| Tier three coach *vs.* tier two coach | 22 | 11–33 | 70 | 57–83 | 24 | 12–36 | 86 | 76–96 |
| Tier three coach *vs.* tier one coach | 22 | 11–33 | 64 | 51–77 | 36 | 23–49 | 82 | 71–93 |
| Tier two coach *vs.* tier one coach | 50 | 36–64 | 92 | 84–100 | 50 | 36–64 | 92 | 84–100 |
| | **Post-contact** | | | | | | | |
| | **Perfect agreement** | | **1 or less** | | **Perfect agreement** | | **1 or less** | |
| | PA±0 | 95% CIs | PA≤1 | 95% CIs | PA±0 | 95% CIs | PA≤1 | 95% CIs |
| | (%) | (%) | (%) | (%) | (%) | (%) | (%) | (%) |
| Tier three coach *vs.* tier two coach | 28 | 16–40 | 62 | 49–75 | 36 | 23–49 | 74 | 62–86 |
| Tier three coach *vs.* tier one coach | 32 | 19–45 | 62 | 49–75 | 38 | 25–51 | 68 | 55–81 |
| Tier two coach *vs.* tier one coach | 38 | 25–51 | 72 | 60–84 | 38 | 25–51 | 72 | 60–84 |

[a]PA: perfect agreement, CIs: Confidence intervals.

post-contact) of tackling. To determine the level of agreement, an analytical goal of a difference of four or less points was set, based on preliminary data [31], to be sufficiently sensitive to identify changes in tackling competency across time. Intra-coach analysis revealed that the tier two and tier one coach met the analytical goal when scoring composite tackling competency. This demonstrates that two coaches of varying experience can consistently score composite tackling competency, with reference to the analytical goal, using the 18-point criteria provided. However, one coach (tier three) demonstrated weaker levels of agreement compared with the other coaches. Further analysis revealed the areas of the tackle that posed the greatest challenges to agreement between coaches, with the more experienced and qualified coach demonstrating the least reliable tackling competency interpretation, which was unanticipated. To establish the level of agreement in the subcomponent analysis, new analytical goals (pre-contact two or less points; contact and post-contact one or less points) were generated from pilot data [31]. Consistent with the composite tackling scoring results, intra-coach analysis revealed the tier two and tier one coach had 100% level of agreement in all subcomponents, in reference to the analytical goals. Whilst the tier three coach showed weaker levels of agreement when scoring the tackling subcomponents, they demonstrated an improved level of agreement from the first to second assessment, which indicates potential learning effects.

In the current study, the tier one and tier two coaches demonstrated the capacity to achieve agreement with a simple familiarisation process, where all participants were correctly interpreted within the tolerance of the analytical goal. That is, the tier one and tier two coach agreed 100% of the time on every assessed tackle within the tolerance of the analytical goal. This novel finding demonstrates the effectiveness of the 18-point criteria and the familiarisation provided for this sub-population of coaches. However, if perfect agreement was required (i.e., in scenarios where only small increases of one point occurred), the highest agreement between the tier three coach and the other two coaches would range between 8% and 16%. In this scenario, the vast majority of players would be disagreed upon. Indeed, such higher tolerances invoked by the more conservative analytical goal would render this test unfeasible for use, since the highest perfect agreement between the two concurring coaches (tier one and two coach) was 32% (95% CI 19% to 45%). Thus, the consistency and magnitude of disagreement between all coaches within the 'perfect agreement' boundary (i.e., no tolerance) indicates that minor competency changes are less likely to be recognised owing to the sensitivity of the tool. This was anticipated based on the complexity of open tackle scoring [15] and it is important to be cognisant of the tool's sensitivity, since expectation of perfect agreement could lead to mis-programming of training sessions, whereby the chosen exercises/drills may be too challenging for the individual's skill level and be ineffective for skill development. If the tool is used appropriately, utilising an empirical analytical goal, then it offers clear advantages for rugby practitioners to assess player's tackling competency and progress them using suitable training drills.

The tier three coach systematically improved their scoring between the two attempts, but still achieved only 72% agreement within the tolerance of the analytical goal. Based on the 95% CIs, their ability to consistently score the total tackling competency could be as low as 60% and no better than 84%. The tier three coach also frequently disagreed with the other two coaches, with agreement no better than 78% (95% CI 67% to 89%). This raises the risk that a qualified, experienced and familiarised user of the current tackling tool could inconsistently identify criteria and do so differently to other users. Whilst some learning effect has been demonstrated, despite familiarisation, it also poses a risk to the validity of the tool if all users are unable to uniformly interpret the tackling criteria. The best inter-rater score between the tier three coach and the other two coaches of 78% (95% CI 67% to 89%) means that a child's potential to improve their tackling competency, as a result of a programme specifically designed to facilitate this, could be disagreed upon by up to 89% and as little as 67%, in reference to the analytical goal. Thus, in the best-case scenario, 11% of player tackling scores would be disagreed upon by coaches of varying experience and qualification within the tolerance of the analytical goal.

The disagreement across coaches may be, in part, due to the more experienced coach (tier three) relying on acquired experiential knowledge, rather than the exact criteria definition to determine whether each criterion is met [36]. Conversely, O'Donoghue [36] suggested that, although definitions for interpretation of skilled actions (performance

indicators in this instance) may be agreed upon prior to independent identification by separate raters, this may be insufficient to facilitate shared understanding of the 'meaning' in an open sports environment. It is certainly possible that definitions will fail to account for all circumstances during tackling events, yet the tier two and tier one coaches of the current study achieved greater levels of agreement, likely by objectively marking tackling competency against the pre-set criteria. This could be mitigated through further coaching observations and many other continued professional development methods, such as reviewing case studies of previous tackle assessments, where the coach can score the tackling events with the tackling criteria and compare to the scores given by other coaches. It is worthwhile conducting further research to evaluate the best ways in which to train coaches to more objectively and consistently score tackling ability.

Subcomponent intra-coach analysis revealed that both the tier two and tier one coaches met the analytical goals for each subcomponent (Table 3). This shows that two coaches of differing levels of experience, can repeatably score sub-component tackling competency. The same coaches who rated similarly in composite scores repeated their level of agreement when assessing subcomponents. Thus, realistic changes in tackling competency in phases of the tackle, occurring as a result of a coaching intervention, could be recognised within the tolerance of the analytical goal. This is encouraging, since it infers that rugby coaches could focus on developing specific skills of the tackle and are sufficiently consistent to re-assess across time to identify tackle competency at a more refined level. However, as with the composite scoring, during the intra-coach assessment, the tier three coach attained agreement up to 80% (95% CI 69% to 81%) within the analytical goal when scoring the post-contact phase, but as low as 64% (95% CI 51% to 77%) scoring pre-contact. This suggests that in the best-case scenario, 23% of player tackling scores, in the pre-contact phase, could be misidentified, but could be up to 49% within the tolerance of the *a-priori* goal. These results, again, highlight that an experienced tier three coach, with the same level of familiarisation to the tool's criteria, may inconsistently score at the subcomponent level. As the tackling tool is designed to inform coaching practice, if subcomponent competency is not reliably scored, this could lead to incorrect training prescription for players, which may inhibit the players skill development and their safety in a collision event.

Inter-coach subcomponent analysis revealed a 98% (95% CI 94% to 100%) level of agreement in pre-contact between the tier two and tier one coach. However, an agreement level of 32% (95% CI 19% to 45%) was revealed if the goal was 'perfect agreement'. This suggests that, in the best-case scenario, with an analytical goal of perfect agreement, coaches would disagree 55% of the time, and worst-case up to 81% of the time. This is also observed in the contact and post-contact phases, where the highest level of agreement within the analytical goal was 92% (95% CI 84% to 100%), in the contact phase, between the tier two and tier one coach, and 74% (95% CI 62% to 86%), in the post-contact phase, between the tier three and tier two coach, respectively. This implies that coaches of differing levels of experience and qualification agree most of the time during the contact phase, with agreement of 100% across all tackles in the best case. However, there remains some risk (16%) of error in assessment in reference to the analytical goal. Also, the best-case scoring error in the post-contact phase is 26%, which could be as large as 38% in reference to the analytical goal between the tier three and tier two coach. This indicates that coaches can repeatably score tackling competency, but the tool may not be sufficiently sensitive to identify if players improve by a single point in the pre-contact phase. Misidentification of a gradual improvement or decline in a child's tackling competency may place them into an incorrect training group and, therefore, being under- or over-challenged for their ability. This infers that if coaches see little-to-no change in a subcomponent, that the child's skill development needs may be misidentified. This could be due to an inconsistent use of the criteria among coaches of differing levels of experience, despite strict instruction to adhere to the given criteria, which has been reported previously [15]. This may be reflective of more experienced coaches relying upon tacit knowledge, a known phenomenon described in the Skill Acquisition Model of Drefyus and Drefyus [37], whereby novices adhere more to structural rules and have less discretionary judgement when performing a task, whereas experts demonstrate more autonomy, no longer relying on guidelines but show a more intuitive understanding and approach tasks

analytically based on tacit knowledge. This is also evidenced among sports scouts, whereby experienced scouts tend to rely on tacit knowledge to inform their decision making, despite working to technical key performance indicators, which can limit the reliability of play identification and call into question a level of bias [38,39]. Nonetheless, it is an important starting point to encourage those assessing tackling technique, particularly in youth rugby, to develop detailed written criteria to guide competency assessment, which will permit consistent independent assessments. Further research should consider how to improve the consistency of experienced coaches to the same levels of the tier two and one coaches in the current study.

### Limitations

A limitation of the present study is the inclusion of a single rater per coaching tier; therefore, conclusions regarding differences in agreement across coaching levels should be interpreted with caution, and future research should include larger samples of raters within each tier.

### Conclusion

The present study has demonstrated that, in the context of evidence-based analytical goals, qualified rugby coaches of varying experience can reach consistent levels of agreement when assessing tackling competency of schoolboys using the Tackle*TEK* Tool. However, this agreement was not uniform across all coaching levels, and we also raise some concerns in the consistency of scoring of a more experienced coach, which was observed in the composite (whole-tackle) and subcomponent level. Given the open nature of the tackling skill, it is plausible that a more experienced coach may rely more upon acquired knowledge, whereas less qualified and experienced coaches appear to adhere to the set criteria more rigidly, thereby producing more consistent scores within and between raters. Nevertheless, the acceptable level (100% agreement within the tolerance of the analytical goal) of tackling competency scoring in both the intra- and inter-coach assessments for the tier one and two coaches indicates that the new Tackle*TEK* Tool can be used by lower and medium level coaches to assess tackling competency among schoolboys, which can be incorporated into a tackling-specific training programme. Applying the Tackle*TEK* Tool across other playing levels, with appropriate analytical goals to investigate reliability, would increase the robustness for general use the general use throughout rugby union. Given the open nature of the tackling skill, further work is required to understand the ways to increase consistency of more experienced coaches and their parity with those of lower qualifications or experience.

### Author contributions

**Conceptualization:** Robert Owen, Mark Waldron.

**Data curation:** Robert Owen.

**Formal analysis:** Robert Owen, Mark Waldron.

**Funding acquisition:** Mark Waldron.

**Investigation:** Robert Owen, Mark Waldron.

**Methodology:** Robert Owen, Mark Waldron.

**Supervision:** Mark Waldron.

**Validation:** Robert Owen, Mark Waldron.

**Visualization:** Robert Owen, Mark Waldron.

**Writing – original draft:** Robert Owen, Mark Waldron.

**Writing – review & editing:** Robert Owen, Camilla J. Knight, Will Page, Jamie Tallent, Liam Kilduff, Mark Waldron.

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
