## [Decision Letter · Decision Letter 0]

3 Jan 2025

Dear Dr. Owen,

We look forward to receiving your revised manuscript.

Kind regards,

Filipe Manuel Clemente, PhD

Academic Editor

PLOS ONE

Journal Requirements:

3. Thank you for stating the following financial disclosure: This work was funded by Lions Sports Academy Ltd (https://lionssports.academy/). The grant (AMS106867) was awarded to MW. The funders facilitated data collection through collaboration with Swansea University.

Lions Sports Academy awarded MW a grant (AMS106867) for the research.

5. We note that you have indicated that there are restrictions to data sharing for this study. For studies involving human research participant data or other sensitive data, we encourage authors to share de-identified or anonymized data. However, when data cannot be publicly shared for ethical reasons, we allow authors to make their data sets available upon request. For information on unacceptable data access restrictions, please see http://journals.plos.org/plosone/s/data-availability#loc-unacceptable-data-access-restrictions.

Reviewers' comments:

Reviewer's Responses to Questions

**Comments to the Author**

1. Is the manuscript technically sound, and do the data support the conclusions?

Reviewer #1: Yes

Reviewer #2: No

2. Has the statistical analysis been performed appropriately and rigorously?

Reviewer #1: Yes

Reviewer #2: Yes

3. Have the authors made all data underlying the findings in their manuscript fully available?

Reviewer #1: Yes

Reviewer #2: No

4. Is the manuscript presented in an intelligible fashion and written in standard English?

Reviewer #1: Yes

Reviewer #2: No

Reviewer #1: The article titled “The Lions Sports Academy TackleTEK Tool: The intra-and inter-coach reliability of assessing tackling competency in rugby union” is extremely well-written, with scientific and methodological rigor. The results are presented clearly and support the discussion. The authors have employed a well-considered statistical approach to assess the intra-and inter-coach reliability in their study of tackling competency using the TackleTEK Tool. By choosing a non-parametric equivalent of the 95% limits of agreement, they have adeptly addressed the challenges associated with analyzing data on nominal and ordinal scales. This choice is commendable as it aligns with the nature of the data, ensuring that the analysis is both appropriate and robust.

Furthermore, it addresses a topic of extreme relevance by presenting an assessment tool that has the potential to identify key points of tackle technique to be worked on, thereby increasing both the performance and the safety of rugby players. Tools for assessing this skill are rare and often difficult to reproduce; therefore, this assessment model can provide coaches with a new option for measuring the tackling skills of their athletes. The present article also paves the way for a series of suggestions for future work, such as training methods for using evaluation tools like this one and the benefits of their long-term use.

My only observation to the authors is that at the end of the introduction, the objectives are presented, but no hypotheses are raised. However, on page 23, line 311, it is mentioned that “the more experienced and qualified coach demonstrated the least reliable tackling competency interpretation, which was unanticipated,” which seems to suggest there was a prior hypothesis that tier 3 coach will be more or equally capable during the evaluation protocol application. Nevertheless, the article is too robust for this to be a problem.

Therefore, my opinion is to approve the present work in its entirety.

I am grateful for the opportunity to contribute with the authors and wish them success in their future research.

Reviewer #2: Dear authors,

Thank you for the opportunity to review this manuscript. The development of youth rugby (skill) practice is an important aim within this research field, indeed. For this reason, I want to first and foremost commend the authors on their efforts to further the development of this research topic. I hope the sport can benefit from their continued contributions.

In the current study manuscript, there are various positive aspects. The overall intent to obtain a reliable measuring tool, examining both intra- and inter-rater reliability, is good practice. The statistical analysis is largely adequate and seems well-executed and reported. Conversely, the in-depth examination of the introduction, methods, and results section, have raised a number of serious concerns regarding the methodological design and development of this study. In addition, I have secondary concerns regarding the clarity with which certain sections of the manuscript are written. In conclusion, the information presented did not lead me to agree with the overall conclusion of this study.

The manuscript would benefit from greater methodological rigour. More attention should go towards precision and contextualisation of the information conveyed, such as pinpointing the particulars of the references used, as opposed to general lump-referencing or using blanket statements without precisely buttressing the argument for which the reference is integrated. The provision of more depth and appropriate nuance regarding the evidence used to frame the current research more precisely will facilitate the reader’s understanding of the current study and highlight the authors’ expertise. Lastly, greater attention should go towards correct and appropriate academic writing, regarding terminology, syntaxing, and the use of appropriate punctuation. These issues convolute the information presented. In my opinion, the manuscript should be restructured and rewritten to optimise the understanding of the actual protocol and its connection to and framing within the prior pilot study.

The main concerns regarding the methodological design and execution of this research study entail:

The efficacy of the original pilot study (during which the tackling skill component of the TackleTEK programme was collected) was assessed using this non-validated tool, which is being investigated post hoc, in the current study, for its reliability.

Based on this original pilot data for which this tool was used, the progress shown in four of eighteen ‘units’ was, seemingly arbitrarily, deemed adequate as the analytical goals for the current study. Apart from the arbitrary nature, I fear these first two points demonstrate a form of circular reasoning.

Although the participants were defined as being the youth players, as this is a reliability study, the raters are arguably the subjects under investigation. The number of raters (3) is limited. In addition, these three raters differ greatly in their perceived coaching expertise, their only reported differentiating characteristic. This heterogeneity in experience level was however not accounted for as a confounder, and practically dismissed in the conclusion with a caveat. I would have liked to see a greater number of raters within each of the three tier-levels to account for this effect and strengthen this study’s generalisability.

Using duplicates of video footage samples (25x2) to inflate the sample size (50), even though from a different angle, is a questionable practice. Especially considering the availability of sufficient original cases (206).

There is a complete lack of detail regarding the context in which the original player data was collected during more than two and a half years. Consequently, there is no way to assess the in-practice standardisation of the protocol.

The extensive familiarisation process undertaken with the raters by the lead researcher raises the question to what degree the outcome was influences by explicitly making the interpretation of the tool’s criteria as uniform as possible; a quality that would be inherent to a valid and reliable measuring tool. The concern is that its ability to be extrapolated to in-practice youth rugby is therefore compromised.

The choice to use multi-angle video footage (with ad libitum rewatch) instead of live assessment of tackling skill also begs the question to which degree these outcomes are generalisable to youth rugby practice, at all levels, which to my understanding is the ultimate aim.

Data restrictions apply and it seems anonymised data is also not available. Therefore, these already limited results are not reproducible.

Regarding the conclusion of this study, within its design, the results show select aspects of reliability, indeed. Notwithstanding, I do not agree that the information presented unambiguously shows that the tackleTEK tool is an overall reliable method for assessing changes in tackling competency. This is largely based on the questionable methodological design issues, as outlined above.

I believe this study may be a worthwhile addition to a thesis, if its contextual limitations are appropriately framed within the overarching research. However, for the reasons discussed, I am afraid I cannot recommend this manuscript for publication in PLOS ONE.

Please find further review details attached.

.

Reviewer #1: **Yes:**Filipe Oliveira BicudoFilipe Oliveira BicudoFilipe Oliveira BicudoFilipe Oliveira Bicudo

Reviewer #2: **Yes:**Dr. Koen WintershovenDr. Koen WintershovenDr. Koen WintershovenDr. Koen Wintershoven

---

## [Author Response · Author response to Decision Letter 1]

15 Aug 2025

Peer review

GENERAL COMMENTS

Dear authors, dear editor,

Thank you for the opportunity to review this manuscript. The development of youth rugby (skill) practice is an important aim within this research field, indeed. For this reason, I want to first and foremost commend the authors on their efforts to further the development of this research topic. I hope the sport can benefit from their continued contributions.

In the current study manuscript, there are various positive aspects. The overall intent to obtain a reliable measuring tool, examining both intra- and inter-rater reliability, is good practice. The statistical analysis is largely adequate and seems well-executed and reported. Conversely, the in-depth examination of the introduction, methods, and results section, have raised a number of serious concerns regarding the methodological design and development of this study. In addition, I have secondary concerns regarding the clarity with which certain sections of the manuscript are written. In conclusion, the information presented did not lead me to agree with the overall conclusion of this study.

The manuscript would benefit from greater methodological rigour. More attention should go towards precision and contextualisation of the information conveyed, such as pinpointing the particulars of the references used, as opposed to general lump-referencing or using blanket statements without precisely buttressing the argument for which the reference is integrated. The provision of more depth and appropriate nuance regarding the evidence used to frame the current research more precisely will facilitate the reader’s understanding of the current study and highlight the authors’ expertise. Lastly, greater attention should go towards correct and appropriate academic writing, regarding terminology, syntaxing, and the use of appropriate punctuation. These issues convolute the information presented. In my opinion, the manuscript should be restructured and rewritten to optimise the understanding of the actual protocol and its connection to and framing within the prior pilot study.

We thank the reviewer for their time and effort on our manuscript and providing us with a detailed review. The reviewer’s suggestions and challenges have helped us strengthen the manuscript throughout with justifications and explanations added through the introduction, methods, and results which support the findings that have been presented.

The main concerns regarding the methodological design and execution of this research study entail:

The efficacy of the original pilot study (during which the tackling skill component of the TackleTEK programme was collected) was assessed using this non-validated tool, which is being investigated post hoc, in the current study, for its reliability.

Thank you for feedback but we would like to address the reviewer’s interpretation of the current work. To the best of our knowledge, despite numerous examples of tackling tests, there is no known validated tool or established process for validating a tool to identify tackling-related skills. There are a number of tests that have established the reliability (in various ways) of their test and frequently report an error statistic (den Hollander et al., 2019; Gabbett, 2008), but without development of analytical goals. Our approach is in-keeping with the notion of a-priori analytical goals (Atkinson & Nevill, 1998), which we have determined based on pilot data, as suggested by the authors of the seminal work (Atkison & Nevill, 1998). Subsequently, we have applied the non-parametric 95% limits of agreement method (Cooper et al., 2007) and adopted the a-priori analytical goal to evaluate the reliability in a meaningful way. This provides readers with an understanding of the error of the test in the context of a realistic analytical goal. We also provide an assessment based on ‘perfect agreement’ to enable the reader to understand the importance of interpreting error (and its acceptability) in cases where the analytical goal may change. This important addition to the literature has not featured in the interpretation of the reviewer, and we ask that further thought is given to the above information.

Based on this original pilot data for which this tool was used, the progress shown in four of eighteen ‘units’ was, seemingly arbitrarily, deemed adequate as the analytical goals for the current study. Apart from the arbitrary nature, I fear these first two points demonstrate a form of circular reasoning.

Thank you for your feedback and the concern you raise of using pilot data. As above, an established framework has been followed that emphasises the setting of clear evidence-based analytical goals (Atkinson & Nevill, 1998). In the current study, four of eighteen units equated to the median improvement of children in the composite tackling score and provided a means to understand whether the potential errors within and between raters would enable the detection of changes of this magnitude. Therefore, it is quite the opposite to the arbitrary selection referred to above by the reviewer, and we direct them to the papers cited in our responses and the current paper, which states where the goal originated and how it was selected. There is no circular reasoning.

Although the participants were defined as being the youth players, as this is a reliability study, the raters are arguably the subjects under investigation. The number of raters (3) is limited. In addition, these three raters differ greatly in their perceived coaching expertise, their only reported differentiating characteristic. This heterogeneity in experience level was however not accounted for as a confounder, and practically dismissed in the conclusion with a caveat. I would have liked to see a greater number of raters within each of the three tier-levels to account for this effect and strengthen this study’s generalisability.

We thank you for your feedback and we agree that this point is arguable. The participants are the youth players, as they were recruited with informed consent and received the instruction of how to perform the standardised tackling protocol, with the raters deemed to be part of the assessment tool. The generalisability of the tool was not being assessed, and the number of raters was determined based upon current research, in which two or three raters are often used to determine the reliability of tackling assessments (den Hollander et al., 2019; Waldron et al., 2014; Gabbett, 2008).

The inclusion of coaches of different coaching levels was intentional and part of the research question, as stated in the manuscript. By definition, a cofounder is a factor that has an influence on the measured variables, and the fact we measured the effects of this by comparing the tool within and between raters demonstrates that this wasn’t overlooked – it was a main feature of the study.

Using duplicates of video footage samples (25x2) to inflate the sample size (50), even though from a different angle, is a questionable practice. Especially considering the availability of sufficient original cases (206).

We are not sure on what basis this is questionable. As the reviewer alluded to above, part of this study is to assess the new test’s (and the raters of the test) ability to rate tackling competency. The interpretation of the rater when using the tool is under investigation and the conditions are the same for all raters. To test the reliability of their scoring whilst incorporating different views/camera angles, this approach is acceptable. Indeed, this provided an opportunity to test rater scoring under altered conditions, which may even strengthen the design.

Our dataset appears to be more than sufficient, given that research in this area has utilised as few as five tackles, by five players, scored from a single angle from video footage (den Hollander et al., 2019) to 20 players who repeated four to six tackles to be scored from single angle (Waldron et al., 2014).

There is a complete lack of detail regarding the context in which the original player data was collected during more than two and a half years. Consequently, there is no way to assess the in-practice standardisation of the protocol.

Thank you for your feedback. The standardisation in practice is achieved through the clear scoring against pre-defined criteria and a structured assessment process, rather than the broader context of the data collection (Hopkins, 2000; Atkinson & Nevill, 1998). That said, we have added the time course in which the data was collected, L121-122: “All data collection was performed during the school term from September to April outdoors, between the hours of 2-4 pm, as part of regular games lessons”, to the manuscript to demonstrate the controlled manner in which the data were collected in a real-world school rugby setting. As the aim of the study was to assess the inter- and intra-reliability, the emphasis is on the robustness of the rating tool itself, not the variability of the external conditions across time. I.e. the videos are the same within and between raters. Therefore, though the protocol was clearly conducted in a controlled manner, the conditions for the raters was consistent and this is what matters. We trust that the explicit description in the methods makes the study easily replicable.

The extensive familiarisation process undertaken with the raters by the lead researcher raises the question to what degree the outcome was influences by explicitly making the interpretation of the tool’s criteria as uniform as possible; a quality that would be inherent to a valid and reliable measuring tool. The concern is that its ability to be extrapolated to in-practice youth rugby is therefore compromised.

Thank you for your feedback regarding the efficacy of the familiarisation process. Rigorous familiarisation procedures are necessary to support the development of robust methods and to improve reliability measurements, which should also be replicated in the field (Hopkins, 2000). Had we not included any familiarisation, then there would be many more criticisms and risks to the study. We are happy that this degree of familiarisation was necessary and view it as a strength of the study.

The choice to use multi-angle video footage (with ad libitum rewatch) instead of live assessment of tackling skill also begs the question to which degree these outcomes are generalisable to youth rugby practice, at all levels, which to my understanding is the ultimate aim.

Thank you for your feedback and concern regarding the use of the tool performed with ad libitum rewatch and it’s use across grassroots rugby union. Important decisions on tackling competency (a very complex skill, involving two people) are extremely challenging in a live environment (Waldron et al., 2014; Gabbett, 2008). This approach allowed for raters to provide a repeatable and comprehensive observational process, controlling potential sources of variability in the live environment, which can affect the measurement of reliability. Additionally, the use of ad libitum rewatching at variable speeds enables raters to review ambiguous moments within a tackle multiple times to reduce the likelihood of errors through misinterpretation, which can occur in live assessments (Spitz et al., 2018). The use of recorded footage is a more responsible way for coaches to focus on the young players performing tackles. If we had chosen live assessments only, there would be many more criticisms of this approach.

Data restrictions apply and it seems anonymised data is also not available. Therefore, these already limited results are not reproducible.

The reviewer can fully replicate the methods of this study and does not need access to results to do so. As per the ethical approval of this research, the raw results are unavailable. The transparent methodological approach can be followed by others and their own evidence-based analytical goals can be determined.

Regarding the conclusion of this study, within its design, the results show select aspects of reliability, indeed. Notwithstanding, I do not agree that the information presented unambiguously shows that the tackleTEK tool is an overall reliable method for assessing changes in tackling competency. This is largely based on the questionable methodological design issues, as outlined above.

We thank you for your feedback; however, based on our responses above, we trust that the reviewer can understand that the link to real-world and context-specific analytical goals has been made. It would be useful in the future to apply the tool to other scenarios where the goals are different, and this would increase the generalizability of its use. The overriding message here is that there is no such ‘black and white’ or ‘yes and no’ conclusion for the reliability of a measurement/test, as this clearly depends upon the use-case, and to conclude anything outside of what we have would clearly be erroneous, despite many other researchers doing so. We have stated this in a future research section of the discussion (L455-457).

L455-457: “Applying the TackleTEK Tool across other playing levels, with appropriate analytical goals, would increase the general use of the tool throughout rugby union.”

I believe this study may be a worthwhile addition to a thesis, if its contextual limitations are appropriately framed within the overarching research. However, for the reasons discussed, I am afraid I cannot recommend this manuscript for publication in PLOS ONE.

We thank you for your candid feedback regarding the manuscript; however, we hope that our responses allay your concerns, whilst demonstrating the importance of this work.

Please find further review details below.

ABSTRACT

Line 25-26: Strangely worded: “...however, it is important to establish the reliability of these tests to understand the capacity to identify change across time.”

Reworded to clarify that previous tackling assessments have focused on talent identification and not been tested specifically for the purpose of monitoring skill development across multiple time points.

L23-25: “Tackling skill tests in rugby aim to evaluate skill competency, but their reliability to detect meaningful changes in tackling competency over time, that align to real-world analytical goals, remains uncertain.

This specific test or all such tests? The tool’s capacity to identify change? Change of what? Be specific, please. I believe you mean to say, you want to be able to confidently (reliably) identify change in tackling competency across time, using this specific test. Therefore, the testing tool needs to be reliable, correct? For instance, an unreliable tool would perhaps be more prone to “identify change”, by its very nature. So too could an invalid testing tool, by not measuring the variable of interest effectively. Please rephrase this sentence.

L23-25 Reworded to clarify the with specific wording to reduce ambiguity.

Line 29-30: “Analytical goals were developed…” This sentence is a filler with no added benefit to understanding the design, as it does not hold any useful information at this point in the manuscript; what analytical goals, why, how were they developed, and by whom? Please frame it so that its importance becomes apparent. Or alternatively leave it out if redundant.

Sentence added to show an objective goal was formed from pilot data using a rigorous iterative process, which is necessary to be able to assess reliability.

L28-309 “Pilot data was used to objectively develop a-priori analytical goals necessary for assessing the reliability of composite and subcomponent scoring.”

Line 30: Subcomponent, no hyphen, is the more common spelling. Both allowed, though.

The hyphen has been removed throughout the manusctipt for the word subcomponent.

Line 33: Tacking missing an “l”.

This has been corrected.

L33: “…tackling…”.

Line 34-35: “the analytical goal of ≤4 points” at this point in the manuscript does not hold much meaning to the reader, as it has not been explained or framed. This statement assumes inside information i

---

## [Decision Letter · Decision Letter 1]

11 Nov 2025

Dear Dr. Owen,

Thank you for submitting your manuscript to PLOS ONE. After careful consideration, we feel that it has merit but does not fully meet PLOS ONE’s publication criteria as it currently stands. Therefore, we invite you to submit a revised version of the manuscript that addresses the points raised during the review process.

We look forward to receiving your revised manuscript.

Kind regards,

Filipe Manuel Clemente, PhD

Academic Editor

PLOS ONE

Journal Requirements:

Additional Editor Comments:

Dear authors

The second reviewer provided a new report with some feedbacks and comments.

Reviewers' comments:

Reviewer's Responses to Questions

**Comments to the Author**

Reviewer #1: All comments have been addressed

Reviewer #2: (No Response)

2. Is the manuscript technically sound, and do the data support the conclusions?

Reviewer #1: Yes

Reviewer #2: Partly

3. Has the statistical analysis been performed appropriately and rigorously?

Reviewer #1: Yes

Reviewer #2: Yes

4. Have the authors made all data underlying the findings in their manuscript fully available?

Reviewer #1: Yes

Reviewer #2: No

5. Is the manuscript presented in an intelligible fashion and written in standard English?

Reviewer #1: Yes

Reviewer #2: Yes

Reviewer #1: The text is well-written, and there have been several improvements since the first revision. However, I point out a few issues that I believe are important to avoid ambiguities or doubts in the interpretation of extremely relevant information regarding the application and validity of the TackleTEK tool.

Abstract:

L44: Since the level of agreement was demonstrated only for tier one and tier two coaches, I suggest stating that TackleTEK can be used specifically by experienced and trained coaches to assess changes in tackling competency.

Video Selection:

L178: The term "healthy participants" is broad and somewhat vague. It would be beneficial to clarify the absence of any previous injuries that could potentially affect the results. Additionally, is there no mention of exclusion criteria? If not, it would be helpful to include this information.

Discussion:

L416-421: In this paragraph, it is noted: “These results, again, highlight that an experienced tier three coach, with the same level of familiarization to the tool’s criteria, may inconsistently score at the subcomponent level. As the tackling tool is designed to inform coaching practice, if subcomponent competency is not reliably scored, this could lead to incorrect training prescriptions for players, which may inhibit the players' skill development and their safety in a collision event.” It is important to emphasize that TackleTEK should be performed by experienced and trained coaches to ensure the quality and reliability of the data obtained.

L454-456: I think it would be useful to suggest that future research could explore comparing the scores obtained during tackles performed in the TackleTEK test with those obtained during real matches. This could help evaluate whether TackleTEK can effectively demonstrate competency in real-world scenarios.

Conclusion:

L460-465: You may want to highlight the importance of adhering to the set criteria rigidly, regardless of the coach's experience level, to ensure the quality and reliability of the data.

Reviewer #2: Extensive work has been done by the authors. Considerable improvments were made. Very commendable. Some clarifications, corrections, and optimisations remain to be done. The main consideration are optimising the statements in the conclusion and the transparancy of the underlying developmental process of the instrument, as well as the availability of the data.

Please refer to the Reviewer's letter.

.

Reviewer #1: **Yes:**Filipe Oliveira BicudoFilipe Oliveira BicudoFilipe Oliveira BicudoFilipe Oliveira Bicudo

Reviewer #2: **Yes:**Dr. Koen WintershovenDr. Koen WintershovenDr. Koen WintershovenDr. Koen Wintershoven

---

## [Author Response · Author response to Decision Letter 2]

25 Feb 2026

PEER REVIEW ROUND 2

The Lions Sports Academy TackleTEK Tool: The intra-and inter-coach reliability of assessing tackling competency in rugby union

INTRODUCTION:

Dear authors,

In the following review, I have critically appraised the responses provided to the first review. Considering the extensive work you have done and the significant improvements this carries with it, I have to sincerely congratulate you on your diligence and the dedication to improve your research. These efforts have resulted in a manuscript of higher quality. In addition to that, I believe it would also be more productive to refrain from using any emotionally loaded language and ad hominem fallacies within your replies.

Notwithstanding the improvements that were made and the clarifications that were provided, there are still some concerns that need to be addressed. Some of them are minor details, other concerns require confirmation or further clarification, and some require more impactful changes. Overall, please consider that this manuscript is closer to publication and simultaneously, the critical nature of this peer review is intended as such. This journal has high standards and good intentions do not automatically lead to immediate successful outcomes. Which does not take away from the authors’ intent.

Please, also recognise that some methodology used in prior published research, used as reference within this manuscript, does not inherently form an adequate basis of support, if the methodology was inadequate or suboptimal to begin with. Consequently, the current research is being reviewed on its own merits, and critically assessed, irrespective of former publications perhaps being less optimal. The aim is to accentuate this manuscript’s strengths while mitigate its limitation and upkeep the scientific rigour for which PLOS ONE is known.

Considering the extent of the reviewed and improved manuscript, in the first section below (p. 2), the authors can find an integral text addressing most pressing issues, as chronologically presented in the former document and addressed by the authors (round 1). In-depth replies to the general comments are provided and may also overlap with and address specific subsequent sections within the manuscript. Responses that were not addressed were found less essential or sufficiently buttressed by the authors at this current stage. Following this in-depth appraisal, the authors can find a section with minor amendments to specific phrases as well.

Overall, well done.

Kind regards.

2nd APPRAISAL GENERAL IN-DEPTH COMMENTS:

RESPONSE TO THE AUTHORS’ REPLIES IN THE FIRST REVIEW

To address the first part of your response; I do not see the purpose of this comment, as it is incorrect and the author seems to know this; on several occasions in this review the author contradicts this statement of there not being any known validated instruments or processes by stating “an established framework has been followed…”, “the recommended process…”, “current process follows the recommended one”, and others alike. In addition, some evidence (references) is provided for it, such as what follows below. Your stance on this matter is therefore inconsistent.

We thank the reviewer for raising this point and acknowledge that our wording may have contributed to this misunderstanding. To clarify, our position is not that there are no established methodological frameworks for assessing reliability. Rather, we state that there is no known validated tackling-specific instrument or gold-standard process for assessing changes in tackling competency over time.

The references cited (e.g., Atkinson & Nevill, 1998; Cooper et al., 2007) describe recommended statistical frameworks for evaluating measurement error and reliability once a tool exists. These frameworks do not constitute validation of a specific tackling assessment, nor do they define technical or biomechanical criteria for competent tackling.

Therefore, there is no inconsistency in stating that (a) no validated tackling competency instrument currently exists, while (b) established and recommended frameworks for assessing reliability were followed in the present study. Indeed, the absence of a validated instrument is precisely why adherence to these methodological frameworks is necessary.

To my knowledge and for your information, the Bangor Rugby Assessment tool is currently the most practically useful and strongest scientifically validated tool with good reliability (doi: 10.3389/fspor.2025.1568302). As this instrument was specifically developed for technical and tactical aptitude in rugby union development pathways, I recommend integrating this into your manuscript.

Thank you for raising this point. We can certainly mention this new addition to the research to the manuscript, but we highlight some key differences below. The Bangor Rugby Assessment Tool was developed and validated for use within rugby union development pathways, with the primary purpose of profiling players’ technical and tactical aptitude to support talent identification and pathway-level decisions, not to inform technical coaching practice. The tool was informed by experts and literature spanning multiple age groups and playing levels, and is intended to provide a broad, game-related assessment of rugby skill performance.

However, the analytical focus of the Bangor Rugby Assessment Tool differs from that of the present study. Specifically, the Bangor Rugby Assessment Tool categorises and evaluates types of tackles and their tactical characteristics within match-like contexts, rather than assessing the technical execution criteria that define a competent tackle. In contrast, the TackleTEK Tool was developed specifically to evaluate criterion-referenced technical behaviours across the pre-contact, contact, and post-contact phases of the tackle, with the explicit aim of informing coaching practice to develop competency. These are apples and oranges. The authors also state that the Bangor Rugby Assessment Tool should be compared to an ‘established measure or method for assessing technical and tactical skills’. This is without reference, but it appears that the authors also accept that this should be compared to something a pre-existing tool.

In addition, the unit of analysis differs between the two approaches. The Bangor Rugby Assessment Tool evaluates multiple technical and tactical components within a broader performance framework and report’s reliability at the level of the overall instrument, not specifically for the tackling component. Important to note, while four rugby coaches and one performance analyst were involved in the development of the overall tool, three of those coaches performed the validity, and only two coaches of differing coaching levels (level 3 and level 4) assessed the reliability of the overall instrument, with no context of coaching experience is specified other than “advanced” and “high-performance”, nothing in regard to years of coaching or level of exposure. Interestingly, within the Bangor Rugby Assessment Tool literature, no specific number of observations is reported to assess reliability. In contrast the present study isolates tackling competency as a discrete context, with full transparency of rater coaching level and experience, and clearly defined sampling (n = 25 participants; repeated observations), rater structure including coaching level, years of coaching experience and level of exposure, and test-retest design to quantify intra-and inter-rater agreement. This is important when considering applicability for comparing the two tools and reliability of the findings.

We also note the language used in the reviewer’s response, which is inadequate when describing reliability. ‘Good’ reliability is meaningless and not actionable. This is a major point of our work that the reviewer must attempt to follow and is apparent in the references used throughout the paper. Therefore, there are two quite major points that the reviewer is not grasping.

Furthermore, as you put forward, den Hollander et al. (2019) provide some backing (with moderate evidence of construct validity) for the current study. However, Gabbett (2008) is questionable as a reliability study, indeed. Moreover, that study holds limited statistical power and carries little weight in terms of evidence of intra- or interrater reliability. Their main aim was not to prove any form of reliability, rather, to investigate the influence of fatigue on tackling technique and relate it to physiological capacity variables.

We thank the reviewer for this clarification and agree that Gabbett (2008) was not designed as a reliability study, nor does it provide strong evidence of intra- or inter-rater reliability. Similarly, while den Hollander et al. (2019) provides moderate evidence of construct validity, it was not cited as evidence of reliability. The inclusion of these studies was intended to acknowledge their practitioner-informed approaches and their contribution to the development of representative tackling assessment environments, rather than to support claims of measurement reliability. Elements of this work have helped develop what we have.

In addition to these studies, the Tackle Ready framework could provide guidance in developing and validating an instrument, such as the one under review. Further to that, parallel research, such as in football (doi: 10.3389/fpsyg.2019.00022) or tennis (doi: 10.3389/fpsyg.2018.02418) can be very useful as a source of methodological guidance.

We thank the reviewer for this suggestion and agree that frameworks such as Tackle Ready, alongside parallel work in other sports, provide valuable methodological guidance for instrument development and validation. These approaches informed the conceptual design of the current assessment; however, formal framework-based validation was beyond the scope of this reliability study and represents an important direction for future research.

Towards the process for validating skill-related competency tools; this process is not fully rigid, but nevertheless fairly well-established, indeed:

1. Conceptualise the initial tool design: thorough literature review (ID key technical or tactical variables,…), expert group feedback (create content validity through consolidating observational criterion, working definitions, etc); drafting tool

2. Validity testing: (face validity), construct validity, criterion validity (compare to ‘gold standard’ or alternatively, objective standard such as expert observation of successful/safe match tackles, e.g.)

3. Reliability testing: inter- and intra-rater. As per the aim of the current investigation.

4. Refinements: ecological validity, practical/logistical utility,… and necessary modifications.

Please look further into the necessary details.

We thank the reviewer for highlighting the need to clarify the developmental basis of the TackleTEK Tool. In response, we have revised the manuscript to explicitly describe the structured workflow underpinning tool development, including literature-informed identification of tackling competency indicators, integration of academic and practitioner perspectives, pilot testing within a grassroots rugby environment, and iterative refinement of the protocol and scoring criteria based on feasibility and clarity for coach interpretation (L95-117). While the present manuscript is not a formal validity study, we recognise that validity testing represents an important next step in the ongoing development of the TackleTEK Tool and have clarified this positioning within the revised manuscript.

I believe that, in the current study and the pilot study, various of these components were targeted. But perhaps not all components were present, designed, or executed to the utmost extent, in my view. If, however, all components were diligently executed, then the communication and transparency regarding its development is lacking; other than a narrative general description in the current study, I have not seen compelling evidence of the developmental workflow of this instrument, specifically, and its (partial) validation (i.e., nr. 1 and especially nr. 2 in the process described above).

We thank the reviewer for this observation and agree that greater transparency regarding the developmental process was warranted. While the development of the TackleTEK Tool involved ongoing discussions between academics and grassroots coaches to refine criteria and protocol design, these discussions were conducted as part of applied programme development and were not formally recorded or analysed as qualitative data. As such, it is not possible to retrospectively present a complete documentary record of all developmental discussions.

In response, we have revised the manuscript to clearly describe the structured workflow that guided tool development and to delineate which components of the broader validation process were addressed within the current study and associated pilot work. We have also clarified that comprehensive formal validity testing remains a vital next step in the continued development of the TackleTEK Tool.

Indeed, the current study aim is to test the instrument’s reliability. I realise that. But considering its reliability (are the measurements consistent?) is part of the greater process of the tool’s testing and evaluation, which is normally preceded by its validity testing (does it measure what it is supposed to measure?), it is perfectly reasonable and logical to have an insight into this component. Otherwise, without evidence of the instrument measuring what it is supposed to measure, its reliability is rendered mute.

We agree with the reviewer’s central premise that reliability testing forms part of a broader process of instrument evaluation and that, in principle, evidence of validity is required to fully interpret the meaning of reliable measurements. We also agree that reliability alone is insufficient to establish whether an instrument measures what it is intended to measure.

In the present case, the decision to focus on reliability was based on the applied context of the TackleTEK Tool and the stage of its development. Prior to formal psychometric validation, the tool was grounded in existing literature on tackling technique, practitioner expertise, and pilot implementation within a representative grassroots environment. These elements provide an initial, implicit basis for content and face validity, although we acknowledge that this does not substitute for formal validity testing.

The aim of the current study was therefore to establish whether coaches could apply the proposed criteria consistently, to an acceptable level of agreement, which is an essential prerequisite for subsequent validity testing and refinement. We have clarified this positioning in the revised manuscript and explicitly acknowledge that comprehensive assessment of face, construct, and criterion validity represents a vital next step in the evaluation of the TackleTEK Tool.

Thank you for adding the reference (31) of the ECSS 2024 Book of Abstracts. This is interesting information in itself. However, its content does not address the above issue. This information does not seem to be available. Can the authors please include or disclose their systematic approach to developing the TackleTEK instrument, other than generically referencing “pilot data”. Preferably, a step-by-step workflow document addressing the design phase and validation subcomponents, as outlined above, would be feasible. This can be done within the manuscript or as an additional document made available alongside this manuscript.

Thank you for your clarification, in response, we have revised the manuscript to explicitly describe the structured development workflow. Specifically, we now outline the step-by-step process undertaken prior to reliability testing (L95-99).

We also clarify that this manuscript is a reliability study and does not constitute a formal validity paper; however, we identify validity testing as an important next step in the instrument’s evaluation pathway.

With that in mind, it is true that the process for developing such an instrument optimally is not a simple undertaking, nor is it fully uniform across the literature. Many studies

---

## [Editor Report · Decision Letter 2]

25 Mar 2026

The Lions Sports Academy TackleTEK Tool: The intra-and inter-coach reliability of assessing tackling competency in rugby union.

PONE-D-24-44137R2

Dear Dr. Owen,

We’re pleased to inform you that your manuscript has been judged scientifically suitable for publication and will be formally accepted for publication once it meets all outstanding technical requirements.

Kind regards,

Filipe Manuel Clemente, PhD

Academic Editor

PLOS One
---

## [Editor Report · Acceptance letter]

PONE-D-24-44137R2

PLOS One

Dear Dr. Owen,

I'm pleased to inform you that your manuscript has been deemed suitable for publication in PLOS One. Congratulations! Your manuscript is now being handed over to our production team.

Kind regards,

on behalf of

Dr. Filipe Manuel Clemente

Academic Editor

PLOS One